# A Deep Learning Approach for Liver and Tumor Segmentation in CT Images Using ResUNet

**DOI:** 10.3390/bioengineering9080368

**Published:** 2022-08-05

**Authors:** Hameedur Rahman, Tanvir Fatima Naik Bukht, Azhar Imran, Junaid Tariq, Shanshan Tu, Abdulkareeem Alzahrani

**Affiliations:** 1Department of Creative Technologies, Faculty of Computing & AI, Air University PAF Complex, Islamabad 44000, Pakistan; 2Faculty of Information Technology, Beijing University of Technology, Beijing 100024, China; 3Department of Computer Science, Air University, PAF Complex, Islamabad 44000, Pakistan; 4Department of Computer Science, National University of Modern Languages (NUML), Rawalpindi Campus, Islamabad 44000, Pakistan; 5Computer Engineering and Science Department, Faculty of Computer Science and Information Technology, Al Baha University, Al Baha 65515, Saudi Arabia

**Keywords:** computed tomography, deep learning, liver segmentation, medical imaging, residual network, tumor segmentation

## Abstract

According to the most recent estimates from global cancer statistics for 2020, liver cancer is the ninth most common cancer in women. Segmenting the liver is difficult, and segmenting the tumor from the liver adds some difficulty. After a sample of liver tissue is taken, imaging tests, such as magnetic resonance imaging (MRI), computer tomography (CT), and ultrasound (US), are used to segment the liver and liver tumor. Due to overlapping intensity and variability in the position and shape of soft tissues, segmentation of the liver and tumor from computed abdominal tomography images based on shade gray or shapes is undesirable. This study proposed a more efficient method for segmenting liver and tumors from CT image volumes using a hybrid ResUNet model, combining the ResNet and UNet models to address this gap. The two overlapping models were primarily used in this study to segment the liver and for region of interest (ROI) assessment. Segmentation of the liver is done to examine the liver with an abdominal CT image volume. The proposed model is based on CT volume slices of patients with liver tumors and evaluated on the public 3D dataset IRCADB01. Based on the experimental analysis, the true value accuracy for liver segmentation was found to be approximately 99.55%, 97.85%, and 98.16%. The authentication rate of the dice coefficient also increased, indicating that the experiment went well and that the model is ready to use for the detection of liver tumors.

## 1. Introduction

The liver is the second largest organ in the body, located on the right side of the abdomen, and weighs about three pounds. The liver has two lobes, right and left, and is in contact with the gallbladder, pancreas, and intestines. Several organs are involved with the liver. Cancer in the liver may be primary (originating from various cells that compose the liver), secondary, or metastatic (caused by cancerous cells from other organs). Malignant hepatocellular carcinoma (HCC) is the most typical primary liver disease among all liver cancers.

The second most common global disease is liver cancer. According to data from the World Health Organization (WHO), it accounted for 8.8 million deaths in 2015, out of which 788,000 deaths were caused by carcinoma [1]. The American Cancer Society (ACS) predicted that around 20,710 new cases in the USA would be diagnosed in the year (29,200 in men and 11,510 in women). Out of these, 28,920 people (19,610 men and 9310 women) died because of primary carcinoma and intrahepatic epithelial canal cancer in 2017 [2]. Carcinoma is more common in the geographical regions of Africa and accounts for more than 600,000 deaths every year [2].

To identify the formation and texture of the liver, radiologists and oncologists use a computed tomography (CT) or magnetic resonance imaging (MRI). In both primary and secondary hepatic tumor cancer, these abnormalities are significant biomarkers for early disease diagnosis, and progression [3]. Usually, the CT volume scan of the liver is understood using semi-manual or manual techniques, but these techniques are costly, time-consuming, subjective, and prone to error. Several calculation methods have been developed to address these issues and improve liver cancer’s diagnostic performance. However, these systems were deficient in segmentation and detection of liver lesions due to several challenges: low contrast between the liver and the neighboring organs, such as liver and tumors of different contrast values; changes in the number of tumors; the size of the tumor being too small; tissue abnormalities; and irregular tumor growth [4]. A new approach is therefore needed to overcome these obstacles.

Earlier studies have had drawbacks, as they employed images from improved magnetic resonance imaging and computed tomography. Furthermore, the CNN technique was only used to assess a few types of liver tumors. In recent years, research has centered on creating a fully automated system for accurate and timely liver tumor prediction while conserving time and energy. The advantage of automatic techniques is that they develop over time as an outcome of their performance and through the integration of various conditions and contributions. Various studies have recently emerged that back up this argument [5,6]. We should try new technologies that have shown promising results in object recognition, image classification, obstacle avoidance, facial recognition, natural language processing, material inspection, and many other applications. These include convolutional neural networks (CNNs). A deep network learns to recognize more complex features by categorizing and combining features from previous layers. This function is known as feature tier, allowing deep learning networks to manage extremely large high-dimensional data, with millions of inputs passing the nonlinear function [7].

Most significantly, the CNN model has been verified to be extremely strong in assessing fluctuating image appearance, which inspires us to apply them to fully automatic liver and tumor segmentation in CT volumes. This study aims to fill a void in the previous research by evaluating the results of a deep learning framework model on a broad dataset of tumor volumes from 3DIRCAD1. We should try a new approach to overcome the above obstacles. Our goal is to create a powerful and robust deep learning model that can perform the task and locate the region of interest (ROI). Each node level in deep learning is trained in a combination of features different from the last layer of the results. Deep learning frameworks can be used for feature classification as well as automated feature extraction.

In conclusion, this study has the following main contributions:We develop a fully automated system for segmenting liver and tumors from CT scan images in a single run.Based on prior studies and their shortcomings, the researchers in this study attempt to achieve 95% mIOU on HCC tumors using VGG and Inception V4 based on the deep learning models. The research technique is intended to improve accuracy and fulfill expectations in the segmentation of liver tumors.We propose a viable method for classifying liver and tumor cells after failing to achieve the desired results with the UNet model. Then, we develop a model that combines both ResNet and UNet, named ResUNet. This deep neural network model utilizes leftover patterns that use escape rather than simple convolutions, resulting in faster testing with few details.We provide a high-level overview of this technology’s results.We provide a general performance summary of this technique, with comparison to a few other fully automated techniques and define a scope for development based on new data and other features.

The rest of the paper is organized as follows. Section 2 presents the literature with regard to existing datasets. The proposed approach is described in Section 3. Section 4 provides an evaluation of ResUNeT. Finally, the limitation of the proposed approach is given in Section 5 followed by conclusion and future work in Section 6.

## 2. Literature Review

Liver segmentation from medical imaging has progressed significantly in medical practice. The objective is to extract knowledge about the human body that has a broad range of applications, including early disease detection and identification of the direction for a proper cure [8]. There are many techniques for processing medical images, which have their advantages and disadvantages. X-ray, molecular imaging, ultrasound, MRI, positron emission tomography, computed tomography, PET-CT, and ultrasonic images are commonly used imaging modalities [9,10]. Hemangioma, focal nodular hyperplasia, adenoma, hepatocellular carcinoma, intrahepatic cholangiocarcinoma, and hepatic metastasis are all diagnosed with CT, US, and MRI [11]. Contrast-enhanced CT imaging is commonly used for survey examinations to rule out the existence of hepatic and extrahepatic metastases and assess the local involvement level since it has a high sensitivity (93%) and specificity (100%). CT scans are widely used to detect liver cancer. Rex and Cantlie first proposed hemilivers and serge as features for manual liver segmentation [12].

However, automated liver and its lesion segmentation remain challenging due to the inconsistent variations between liver and lesion tissue caused by various acquisition methods, contrast agents, contrast enhancement levels, and scanner resolutions [13]. Compared to previous examinations, convolution neural networks (CNN) can help in deep learning with regard to liver lesions [7]. CNN’s performance is the best and, in some cases, has surpassed the knowledge of human radiology. In the medical domain, CNNs have been widely used to detect various tumor forms over the last few years.

The researchers used convolutional neural networks and other deep learning systems to incorporate ideas about liver tumor diagnosis. Both supervised and unsupervised classification is possible. The feature sets are grouped into predefined groups in a supervised system, while they are allocated to undefined classes in the unsupervised method. To train and evaluate the output of a classifier always requires training and testing data [14]. The data for medical image training is normally collected from one or more experts who have assigned labels to a set of objects. They distinguished between regular and abnormal liver cells in a dataset that included 79 H&E-stained liver tissue WSIs of hepatocellular carcinoma (HCC), 48 of which were HCC tissue and 31 of which were normal tissue [15]. The authors suggest a mechanism for detecting differences in normal neural networks and in CNN, using a high and low enlargement in the cell map and surface structure. The results showed a 91% probability of correct liver HCC tumor detection using CNN [15].

Deep learning (DL) techniques have excellent learning abilities [16]. Deep learning models such as the convolutional neural networks (CNNs), stacked automatic encoder (SAE), deep belief network (DBN), and deep Boltzmann machine (DBM) have been implemented [3,17]. Deep learning models are superior in terms of accuracy. After all, finding a suitable training dataset, which ought to be large and constructed by specialists, remains a significant challenge. The literature revealed that DL-based models for liver tumor detection have attained 94% accuracy.The CNN model comes in a variety of architectures [18,19], including AlexNet, VGGNet, ResNet, and others. While [12,20] used the VGG16 architecture in their research, other research [3,12,21,22,23] has employed the two-dimensional (2D) UNet, which is primarily used for splitting up medical images [24].

On the other side, one research report classified benign and malignant tumors. They used an enhanced contrast ultrasound (ECUS) dataset in video form, separated into two subgroups: 20% test data and 80% training data. Another study [15] suggested using CNN to minimize the number of liver tomography images required while reducing the time and money spent on maintenance. The model has been improved, and the potential has been developed to distinguish between benign and malignant tumors using a combination of deep learning and CNN. The researchers used a training dataset of 55,536 photos from the 2013 database and 100 liver images from the 2016 data for testing [7]. The experiments were repeated five times, and the results showed an accuracy of 92%.

In a similar study [12], researchers used computed tomography images and a convolutional neural network to compare various types of liver tumors. The researchers calculated the probability by segmenting each pixel using special segmentation algorithms and a deep convolutional neural network. They used layer wrap to minimize functions and eliminated three-dimensional variation by grouping layers into characteristics and classifying tumors based on a fully connected layer. Another study [13] was performed with 2.5D computed tomography images to segment liver lesions using a deep convolutional neural network using Res-Net and virtual UNet. The scholar used a working model design created in 2.5D instead of 3D. An NVIDIA (Santa Clara, CA, USA) Titan XGPU with 12 GB of memory and 3584 cores were used for training and testing for four days in a row.

Rectified linear units (ReLU) have become one of the most important actuation functions in deep learning and machine learning. The rectified linear activation function is a piece-wise linear function that outputs the input directly if it is positive. Otherwise, it outputs null or 0. It has become the norm trigger function for many types of neural networks because a model that uses it is easier to train and often performs better.

Much research has been performed on liver and liver tumor segmentation using semi-automatic, automatic, and manual techniques. Although manual segmentation varies in different segments or parts of the liver, hemi-liver, or vessels, automatic and semi-automatic methods focus on different algorithms to enable tumor or liver segmentation from medical images with more, less, or no user intervention [25]. This section provides a comprehensive literature review of liver and tumor segmentation based on state-of-the-art deep learning techniques.

## 3. Method

This section focuses on the steps used in implementing the liver and tumor segmentation using the hybrid deep learning network ResUNet. The preprocessing, feature extraction, classification, and segmentation are all part of the proposed method’s pattern in the deep learning network scheme. Figure 1 shows the overall framework of the proposed model for liver and tumor segmentation.

### 3.1. Dataset

The 3D-IRCADb-1 (http://www.ircad.fr/research/3dircadb, accessed on: 25 January 2022) dataset consists of three-dimensional (3D) CT images of patients, which are well ordered and made publicly available by the IRCAD. Each image has a width and height of 512 × 512 pixels. The depth of each patient’s slice, or the number of slices, varies between 74 and 260 for overall 2800 slices. These self-contained 3D CT scans of ten men and women each represent 75% positive cases. DICOM-formatted patient images, labeled images, and mask images are provided as data for the segmentation process in the preprocessing section. Couinaud segmentation [26] reveals the location of tumor volumes, highlighting the key challenges of using software to segment the liver [27].

### 3.2. CT and MRI Images Preprocessing

The proposed method is employed to extract useful segments from liver tumor images. Data augmentation, preprocessing, and CNN is used to diagnose the liver and identify tumors in the surrounding organs. In the preprocessing phase of CT images, Hounsfield unit values in the range of −100 to 400 are passed on, neglecting the adjacent organs. Then, histogram equalization is applied to the image to increase the contrast. Finally, some data from the magnification steps are used to increase the data and teach the desired invariant properties, such as translation, rotation, deformation, elasticity, and the addition of the Gaussian noise standard deviation, as shown in Figure 2.

Every medical image analysis system uses image preprocessing to enhance the quality of the raw input image. This entails noise reduction, enhancement, normalization, and standardization techniques, along with other things. As defining blocks and feature extraction depend on image quality, preprocessing is important to achieve the other steps involved. The normalization and distributing procedures adjust the image’s values and reduce the spectrum, making it easier to improve the classifier. Noise reduction improves image screen resolution and eliminates unnecessary qualities in the image, making other processing tasks, including edge detection, segmentation, and compression, more efficient. Two approaches for eliminating noise from current medical images are the spatial domain approach and the spectral domain approach. Examples of spatial domain approaches are mean filtering, adaptive mean filtering, order-statistic filtering, adaptive weighted median filtering, maximum a posteriori filtering, nonlinear diffusion, geometric filtering, and so on [28]. The mean value of its neighbors replaces each pixel in mean filtering. It gives the picture a smoothing and blurring effect. The adaptive mean filtering technique uses local image statistics such as mean, variance, and correlation to detect and maintain edges and features [29]. Noise is reduced by using a local mean value to replace the original value. This filter adapts to the image’s properties locally, and aids in the selective removal of noise from various areas of the image [25]. Compared to the mean filter, the median filter is an order-statistic filter that creates less blur and preserves edge sharpness. By maximizing the Bayes theorem, an unobserved signal and a maximum a posteriori filter are used to estimate the values [30]. Curvelets may also be used to eliminate noise in medical images. Curvelet transform is a multi-scale conversion with scale and position parameters and indexed frame elements [31]. The results obtained after HU windowing and ResUNet segmentation are illustrated in Figure 3.

Histogram equalization is one of the image processing techniques used to enhance the contrast between the liver and its neighboring organs for more visibility or understanding. This made it easier to define the segmentation of the liver tumor. Histogram equalization is illustrated with before and after CT images in Figure 4. Each CT image slice dataset has its own tumor volumes and liver masks collection.

### 3.3. Data Augmentation

Data augmentation is mainly used to augment data size by performing various steps such as rotating, shifting, skewing, zooming, merging, and so on. This study primarily focused on image, mask reflection, and rotation. Figure 3 depicts the mask image for tumor detection, while Figure 4 shows tumor detection from the mask. Figure 5 explores the merger of final masks from both tumors.

#### 3.3.1. Feature Extraction and Selection

Several features can be derived from medical images, but texture-based features are used for training a classifier or automatic liver or tumor segmentation. Texture analysis provides a wealth of visual data and is an essential part of image analysis [32]. One of the most commonly used statistics is the gray level co-occurrence matrix (GLCM), which is based on second-order statistics of grayscale image histograms [33].

#### 3.3.2. Feature Selection and Merging

All the different masks were combined to improve training and data augmentation. Since the IRCADB01 3D dataset comprises tumor masks for each tumor alone, we had to combine all the masks into one mask to make it easier to train and expand the data.

#### 3.3.3. Reflection Image and Mask

Other researchers have performed liver masks, tumor mask reflections, and adjustment of the Y-axis motion of each slice to improve the dataset’s training efficiency. Figure 6 demonstrates the slice reflection before and after, while Figure 7 depicts the mask before and after reflection.

#### 3.3.4. Rotation image and mask

We rotate each slice containing a tumor along with the tumor mask and liver mask to raise the number of slices, as shown in Figure 8 and Figure 9.

### 3.4. Defining Region of Interest (ROI)

Usually, simple image segmentation is used to determine ROI in medical images. Thresholding, region increasing and boundary monitoring, classifier methods, deformable models, and atlas-guided methods are some popular methods for defining ROI. The method of separating the target area or object of interest from the entire context in preparation for feature extraction is known as region of interest (ROI). Manual, semi-automatic, and automatic processes can be used to define ROI in medical photos. It normally divides pixels into two groups, one for pixels with a specific range of intensity and the other for a wider range of intensity. While it is an easy and effective method, it has its drawbacks, including the inability to account for spatial image characteristics, noise sensitivity, intensity, and inhomogeneity [34,35].

Labels are used in classifier methods to divide the feature space into different classes based on tissue or anatomical area. Supervised classifiers use manual segmentation data as training data, which are then used as a guide for automated segmentation of new data. Unsupervised classifiers use clustering methods that perform the same functions as supervised classifiers without requiring training data. Regions are extracted using deformable models. Deformable models extract area boundaries using a closed parametric surface that changes or deforms in response to the model’s internal force and the image’s external force [36]. There are two types of deformable models: metric deformable models and geometric deformable models. An atlas of anatomy is used to segment organs using knowledge about the anatomy of interest. This approach seems sufficient if the structures are consistent across slices [37].

We used a UNet to segment tumors in the liver, but we obtained poor results, so we tried to segment tumors with ResUNet. This was trained with CT scans of the liver after extracting the ROI from the first CNN along with the cancer masks. Examples are included in Figure 10 and Figure 11.

## 4. Evaluation with ResUNeT

ResUNet was equipped to locate the region of interest (ROI) from the nearby organs using CT scans and liver masks. After separating the ROI and training on CT scans of the liver, the ResUNet was used to segment tumors in the liver. ResUNet swaps convolutional patches for remaining sections, combining the advantages of both models. The deep learning preparation is simple, with residuals in each block, and eliminates relations between the network’s weak and high levels. It also contributes to having limited trainable parameters in each remaining unit. Deep learning preparations are simple to use with residuals in each block and eliminate the relationship between weak and high levels of the network. It also contributes to the limited training parameters in each remaining unit. Figure 12 shows the ResUNet architecture.

A CNN is a feature extractor that performs well in contrast to the other texture extractor features. Compared to other complex textural approaches, the features of the model extracted with a convolutional network and CNNs can take time for training. In classification tasks, CNNs have been shown to be successful [16]. Firstly, the CNN reads data or performs augmentation of the data to train the algorithm, and then it combines both. As discussed earlier, deep learning models will encode and decode the CT images for better segmentation, so the ResUNet comprises three different routes:Encoding route: converts the input into an accurate recognition.Decoding route: reverses the encoding and categorizes the representation pixel by pixel.Bridge processing: joins the two routes.

ResNet, on the other hand, is a simplified version of residual blocks that uses artificial neural networks [13]. In residual blocks, the skip connections principle simplifies and accelerates the deep learning process in complex networks [38]. In contrast, the hybrid ResUNet allows comprehensive standby of convolutional blocks [15].

### 4.1. Segmentation Process of Liver and Liver Tumor

Liver segmentation is the process of segmenting a medical image (CT, MRI, or US) into liver parenchyma and non-liver parenchyma regions. Statistical shape models, graph cuts, clustering, deformable models, area expanding, a level range, thresholding, active contour, support vector machine (SVM), neural network (NN), and other methods are used to segment the liver. Many fully connected networks have recently been developed, and they appear to be promising. Still, they require a large amount of training data and a high-speed processor, making them computationally costly. By studying the homogeneity function of the region, Pohle et al. proposed an adaptive region rising method to segment the liver [39] automatically. Since this approach is based on homogeneity parameters of the tissue, it performs under-segmentation when the target is non-uniform.

Suzuki et al. used fast-marching level collection and geodesic active contour to create a completely automatic system for calculating liver volume [40,41,42,43] using the segmentation outcome of one slice as the original segmentation for another slice. Quick marching and mathematical morphology, as well as static, were used to create a fully automated system for liver segmentation with graph cut. Quick marching and mathematical morphology, statistic adaptive threshold initialization, and k-means clustering were used to create a fully automated system for liver segmentation with graph cut [3,44,45]. Erdt et al. proposed a new SSM based on local shape priors combined with constraints directly derived from the model’s current curvature for fully automatic CT liver segmentation [37]. Figure 13 presents the implementation framework of deep learning neural network on liver and liver tumor segmentation. It involves image preprocessing steps such as noise reduction, standardization, and normalization techniques to enhance the image quality. Data augmentation operations (i.e., reflect the image, rotate the image, and mask) are performed to augment the training samples. ResUNet was prepared to find the region of interest (ROI) from the nearby organs using CT scans. After separating the ROI and training on CT scans of the liver, the ResUNet was used to segment tumors in the liver.

Furthermore, it is important to segment the tumor for any surgical procedure. At various stages of liver cancer, accurate and precise location and shape of a tumor are needed for a better cure plan. Accurate segmentation allows us to monitor the therapy’s progress over time. Based on deep learning classifiers and models, various semi-automatic and automatic techniques for liver tumor segmentation have been suggested. These characteristics were then used to train a Hopfield neural network to classify organs. Only one picture was used for the process, and the result was very disappointing [46] with the suggested segmentation algorithm for a liver image, which combined multi-layer perceptron NN and fuzzy-k-means. A few implementations of a fully linked network to segment the liver and liver tumors have been published recently [47,48,49]. A detect before extract system was proposed by Chen et al. to locate the liver boundary [50] automatically. Deep neural networks (DNNs) are a form of neural network (NN) that have more than one hidden layer or more than three layers (input and output) [27].

Each layer of nodes in deep learning is trained on a different set of features from the previous layer’s output [51,52]. The deep network is taught to recognize more complex features as it goes deeper by aggregating and recombining features from previous layers [53]. This skill is known as feature hierarchy, and it allows deep learning networks to manage extremely large high-dimensional data with billions of parameters passing the nonlinear function [7,53]. DNN can be used for feature classification as well as automated feature extraction. The DNN classifier is trained on labeled data before being applied to unlabeled, unstructured data, allowing it to process much larger datasets. Aside from that, other researchers’ approaches addressed here have only used a single-scale high-magnification patch with a cell-level information-based pattern. The tumor-based cells could not be detected absolutely. Identifying normal and abnormal cells in the liver was a challenging chore that necessitated a thorough examination of the layers of 3D images.

These published studies provide several conclusions. First, as opposed to semi-automatic methods, automatic systems do not degrade segmentation efficiency. In addition, severity alone does not seem adequate for segmenting lesions other than metastases. Following this, a liver envelope appears necessary for segmenting liver tumors, particularly for automatic approaches. Finally, in the case of the liver, deep learning and machine learning methods work well. Previous research has shown that segmentation is best achieved when restricted to the liver, particularly for automatic segmentation. Deep learning is important in segmenting liver tumors, particularly when texture changes differentiate lesions and you have perfect knowledge about the types of liver lesions.

Moreover, deep learning techniques are frequently employed to segment liver tumors. When using texture features, deep learning is particularly useful because the choice and combination of these features are challenging in supervised data schemes. All methods rely on texture features and are limited to an ROI based on the use of deep learning techniques. Deep learning techniques are used in all methods that rely on texture properties but are limited to an ROI based on the injury the user indicates, as this is the best option.

This article attempts to fill the gaps left by previous research. The convolutional network architecture has been used in previous studies to facilitate the use of multiple magnifications while providing information about the cellular structure for low-magnification patches. Based on earlier research, the authors of this study are trying to achieve 91% mIOU in HCC tumors by using convolutional-network-based VGG and Inception V4. The ResNet architecture is a categorized step of the encoder-decoder of the layers, which forms a deep convolution encoder-decoder. The proposed architecture was tested on a typical liver computed tomography dataset or a tumor volume in the training process. The research technique aims to improve accuracy and meet expectations in diagnosing liver tumors. They accompanied another model to get the best results using a UNet model.

To get the best results with a UNet model, the authors created ResUNet, which uses jumps instead of the traditional turns used by traditional networks, allowing faster preparation with less data.

### 4.2. Final Results

All patients were examined with CT images. Diagnostic results are analyzed in terms of sensitivity, accuracy, error rate, and specificity obtained using the values of true positive (TP), true negative (TN), false positive (FP), and false negative (FN). The formulation of these evaluation parameters is shown in the following equations:$$Sensitivity=\frac{TP}{TP+FN}$$
(1)$$=\frac{TP}{Diseased}$$

Probability that the test will correctly recognize a patient who has the disease:$$Specificity=\frac{TN}{TN+FP}$$
(2)$$=\frac{TN}{NoDiseased}$$

Probability that the test will correctly recognize a patient who has the disease:(3)$$Accuracy\left(Acc\right)=\frac{TP+TN}{TP+FP+TN+FN}$$

Accuracy provides general information about how many samples are misclassified:(4)$$ErrorRate=1-Acc=\frac{FP+FN}{TP+TN+FP+FN}$$

Intersection over junction (IoU) is the amount of classified pixels relative to the junction of what is expected and the original value from the same class. The mIoU represents the average between the IoU of fragmented items and the rest of the samples from the test dataset. It can be written as follows:(5)$$IoU=\frac{TP}{TP+FP+FN}$$

Due to the overfitting of the data, validation loss is high after five intervals, and the validation dice coefficient decreases. After using the ResUNet model to segment the liver, the results are shown in Figure 14, Figure 15, Figure 16, Figure 17, Figure 18 and Figure 19. Finally, Figure 20 represents the confusion matrix from the predicted value after tumor segmentation.

Table 1 shows the training progress of the ResUNet model for liver segmentation over 20 epochs. Table 2 shows the results of tumor segmentation and the training progress of the ResUNet model over 50 epochs. The other assessment measurements, which incorporate accuracy and SVD, were additionally determined. The SVD shows the contrast between the real and predicted masks. The proposed model achieved an accuracy of 99% with an SVD score of 0.22, which was the lowest contrast between the actual and predicted masks, as displayed in Table 3. The justification for the higher accuracy was class unevenness. To examine CT images, more pixels have been placed in a foundation class, where the probability of the presence of a tumor is very low. Consequently, the accuracy esteem is biased toward the background class since accuracy counts all classes’ complete numbers of TP, FP, TN, and FN.

## 5. Limitations of the Proposed Approach

We acknowledge that the sample dataset size of the present study is an obvious limitation that prevents us from generalizing results. Despite the Res-UNet producing very promising results, there are a few limitations. We may get around these limitations by planning for more epochs, using more data, using different datasets, or using different preprocessing strategies. The findings will be summarized using some examples first, followed by a discussion of the general case. A collection of images with various types of tumors were subjected to straight segmentation using a classification function. Three slices were chosen to demonstrate segmentation accuracy by contrasting the automatic segmentation with the ground truth.

In addition, a limited-sample-size neural network is considered a risk. Therefore, the results presented should be interpreted with caution, and future research should be carried out to increase the sample size to confirm sufficient support of the results shown. In addition, deep learning techniques are often used to segment liver tumors. Deep learning is particularly useful when using texture features because selecting and combining these features poses a challenge for monitored data schemes. All methods are based on characteristics of texture. Deep learning techniques are used in all methods based on the properties of the texture, but are limited to an ROI based on the injury the user indicates, as this is the best option.

## 6. Conclusions and Future Work

This paper presents the use of a deep learning model for tumor and liver segmentation in CT images. As a result, the hybrid ResUNet is significantly more effective in terms of training time, memory usage, and accuracy as compared to baseline methods. The binary segmentation by classification layout was created to make processing medical images easier. The basic 3D-IRCADB1 dataset was used to train and evaluate the proposed model. The proposed technique properly identifies maximum tumor areas, with a tumor classification accuracy of over 98%. However, after reviewing the data, it was discovered that there were only a small number of false positives, which can be improved by false positive filters and training the model on a bigger dataset.

ResUNet delivered excellent results in terms of diagnosing quickly and efficiently. As we can see from the results, deep learning neural networks assisted us in achieving our goals and are possibly the best tool for dividing liver tumors. They can also be tried with tumors other than liver tumors, as the ResUNet showed promising results. The ResUNet model’s performance can be increased by using more datasets and different preprocessing techniques. It can aid in the diagnosis of liver tumors, with 99.9% precision. The rate of the authentication of DC also increased, suggesting that the experiment went well and that the model is ready for use in detecting liver tumors. In future studies, we plan to explore a new deep learning model at a further level to improve tumor localization accuracy, lower the FN rate, and increase the IoU metric.

## Figures and Tables

**Figure 1 bioengineering-09-00368-f001:**
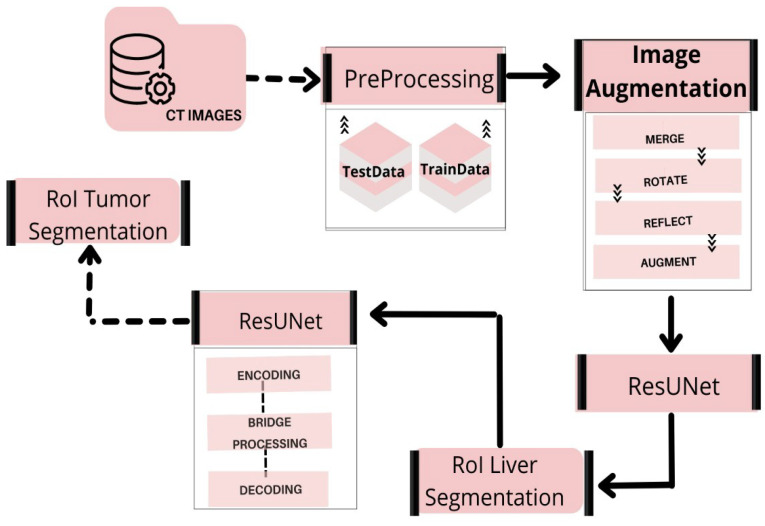
Overview of our proposed workflow.

**Figure 2 bioengineering-09-00368-f002:**
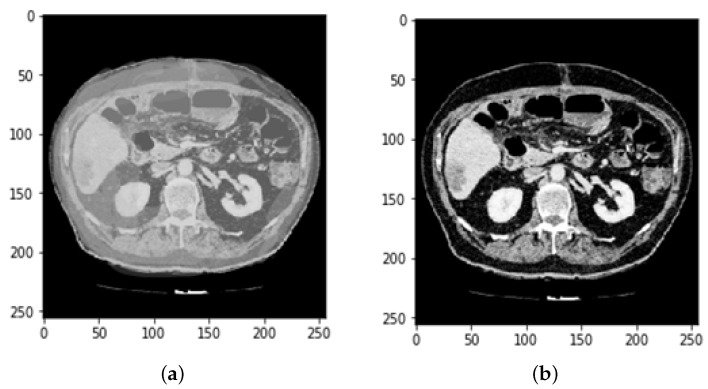
Preprocessing state of CT image (**a**) before Hounsfield unit windowing and (**b**) after HU windowing.

**Figure 3 bioengineering-09-00368-f003:**
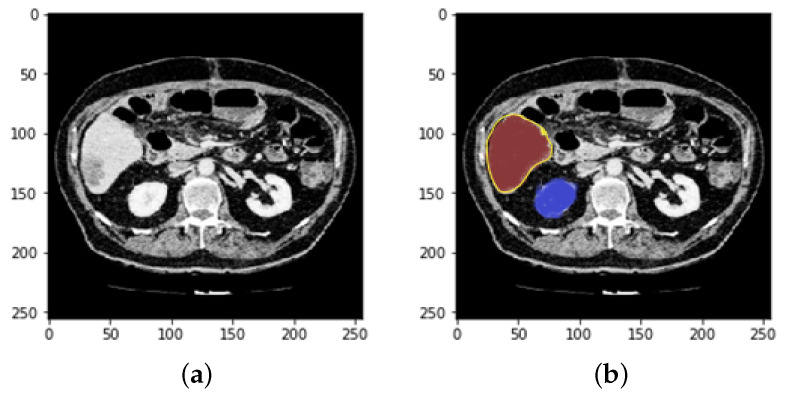
Prepocessing state of CT image (**a**) after HU windowing and (**b**) using ResUNet Segmentation.

**Figure 4 bioengineering-09-00368-f004:**
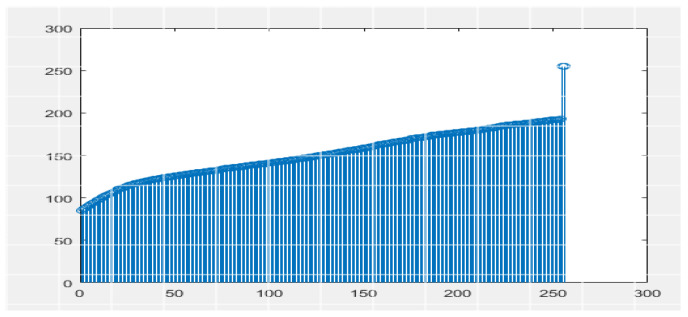
Preprocessing state of CT image by applying histogram equalization.

**Figure 5 bioengineering-09-00368-f005:**
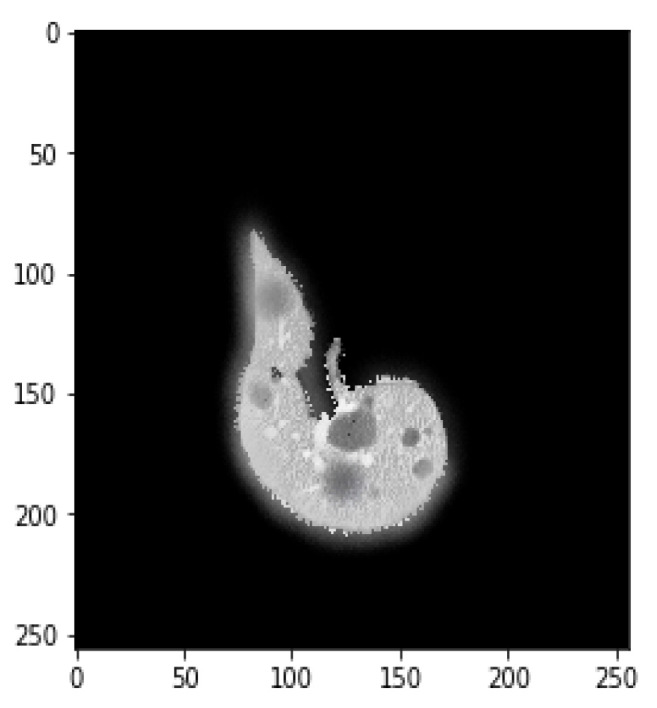
Mask image for tumor detection.

**Figure 6 bioengineering-09-00368-f006:**
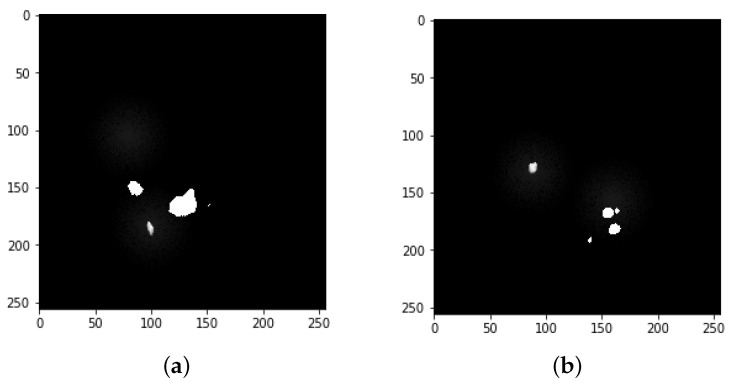
Image mask samples: both (**a**,**b**) show the detected tumors from the mask.

**Figure 7 bioengineering-09-00368-f007:**
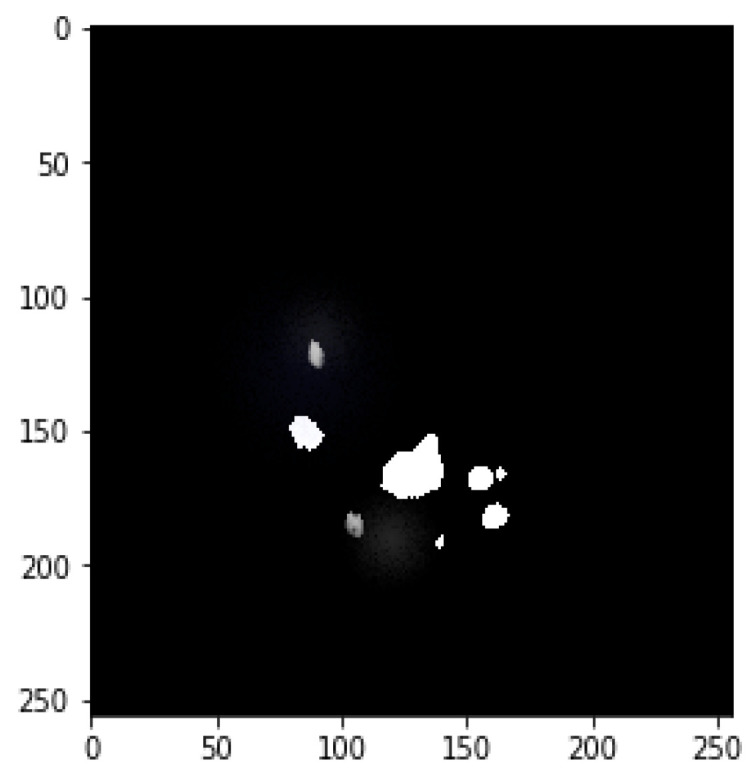
Final mask merged from both tumors.

**Figure 8 bioengineering-09-00368-f008:**
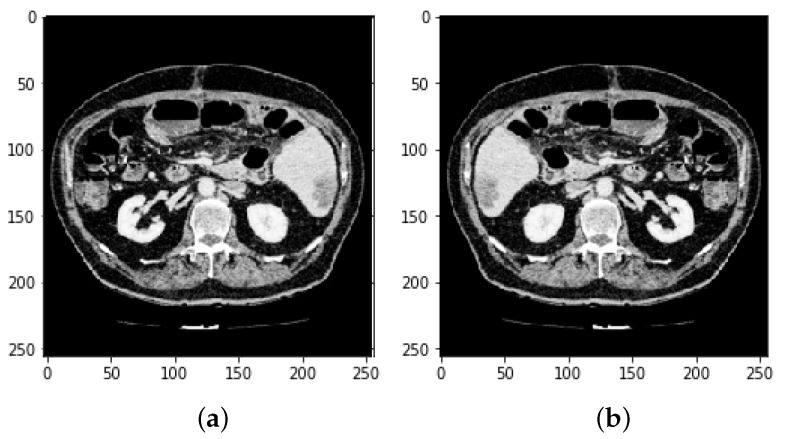
(Image slices **a**) before reflection and (**b**) after reflection.

**Figure 9 bioengineering-09-00368-f009:**
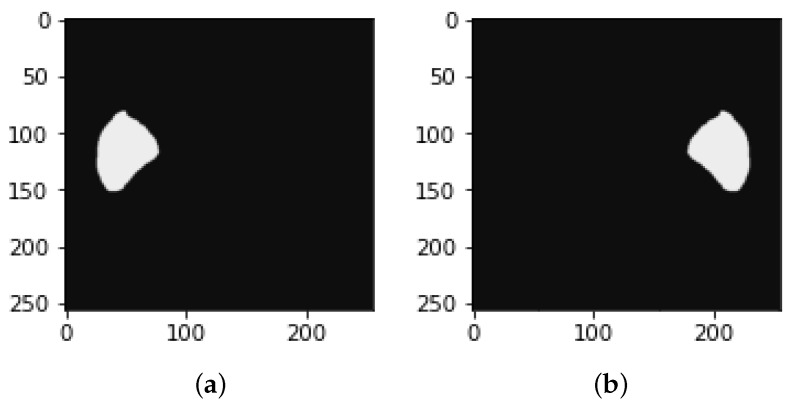
Image of mask (**a**) before reflection and (**b**) after reflection.

**Figure 10 bioengineering-09-00368-f010:**
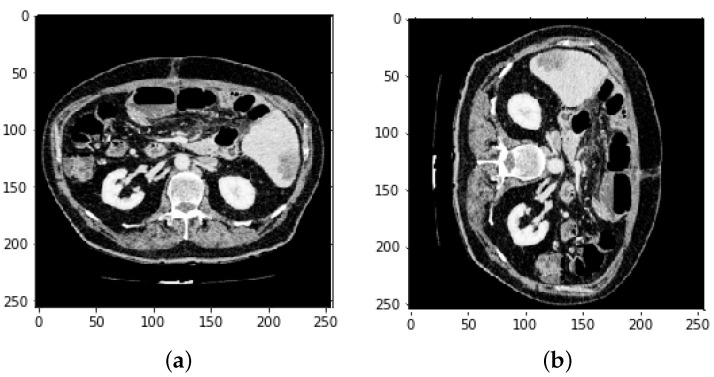
Image slices (**a**) before rotation and (**b**) after rotation by 90${}^{\circ }$.

**Figure 11 bioengineering-09-00368-f011:**
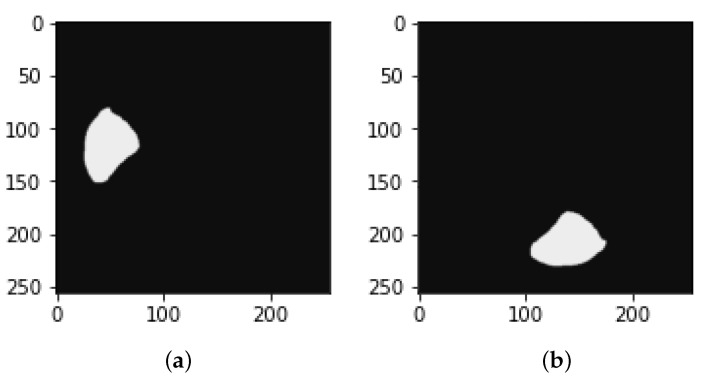
Image of mask (**a**) before rotation by 90${}^{\circ }$ and (**b**) after rotation by 90${}^{\circ }$.

**Figure 12 bioengineering-09-00368-f012:**
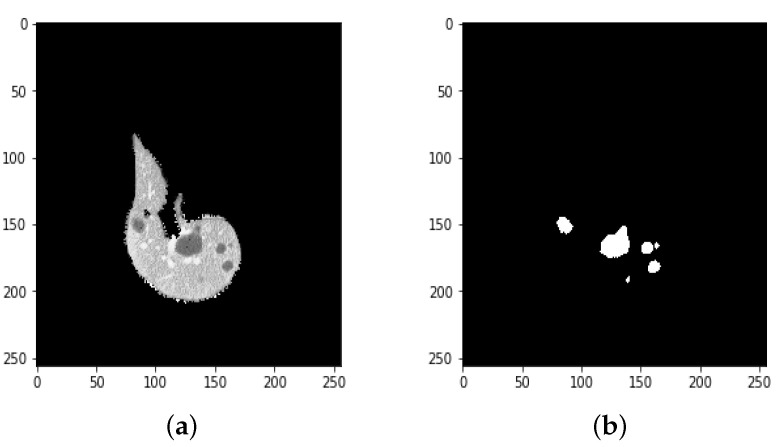
(**a**) A ground truth tumor image and (**b**) the resulting tumor segmentation.

**Figure 13 bioengineering-09-00368-f013:**
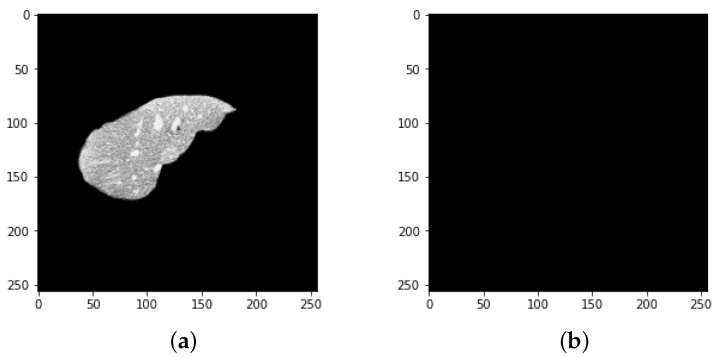
Results of the proposed model (**a**) liver segmentation and (**b**) tumor segmentation (without tumor).

**Figure 14 bioengineering-09-00368-f014:**
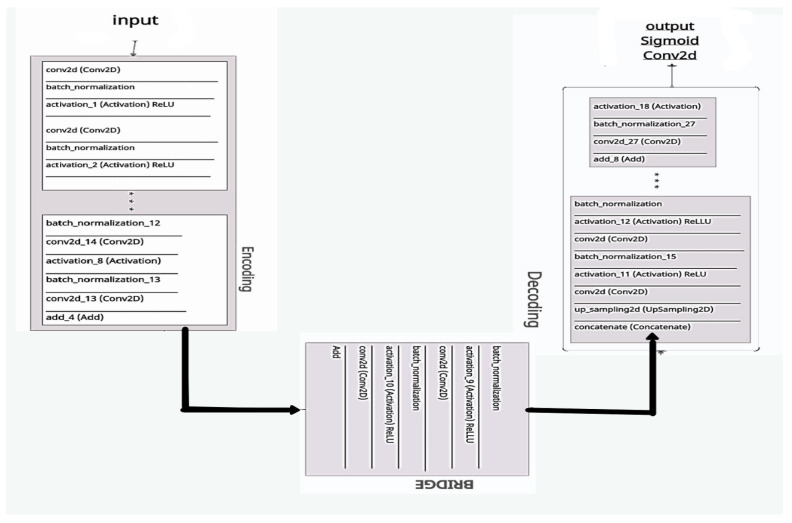
The proposed ResUNet architecture.

**Figure 15 bioengineering-09-00368-f015:**
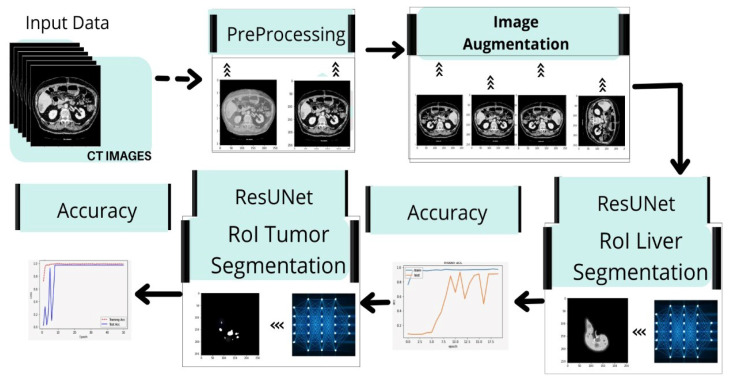
Implementation framework of deep learning neural network on liver and liver tumor segmentation.

**Figure 16 bioengineering-09-00368-f016:**
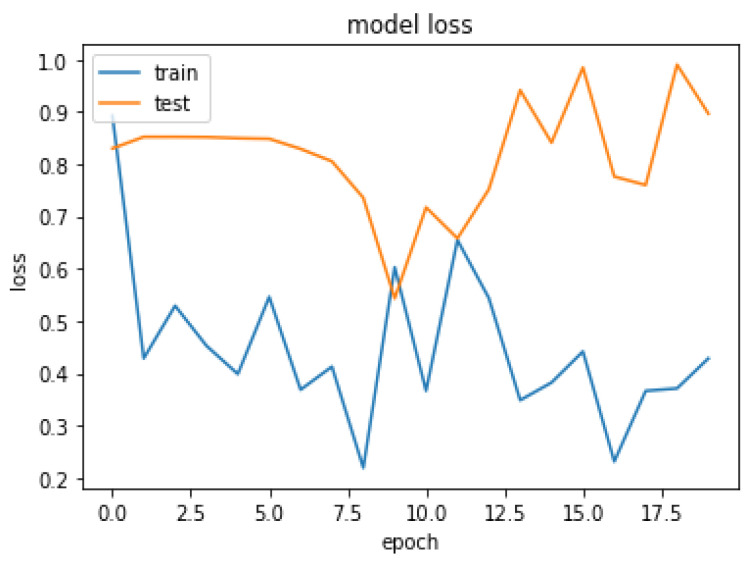
Loss over 20 epochs during the model ResUNet’s training progress for liver segmentation intervals.

**Figure 17 bioengineering-09-00368-f017:**
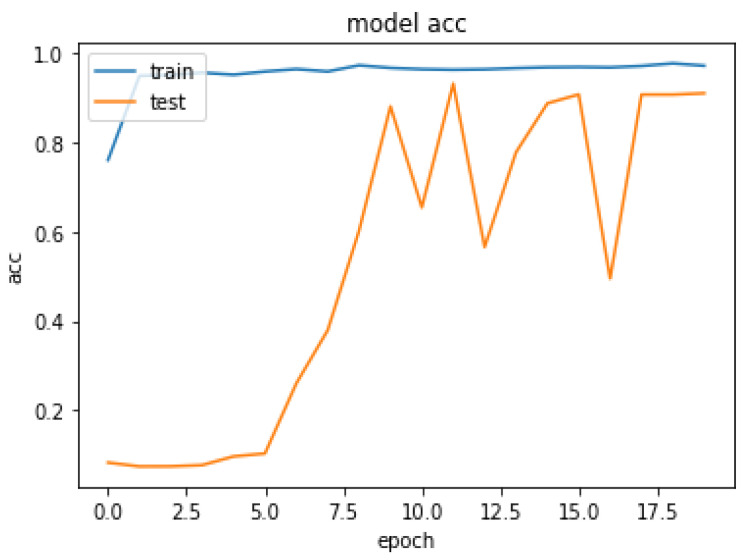
Acc over 20 epochs of the model ResUNet’s training progress for liver segmentation intervals.

**Figure 18 bioengineering-09-00368-f018:**
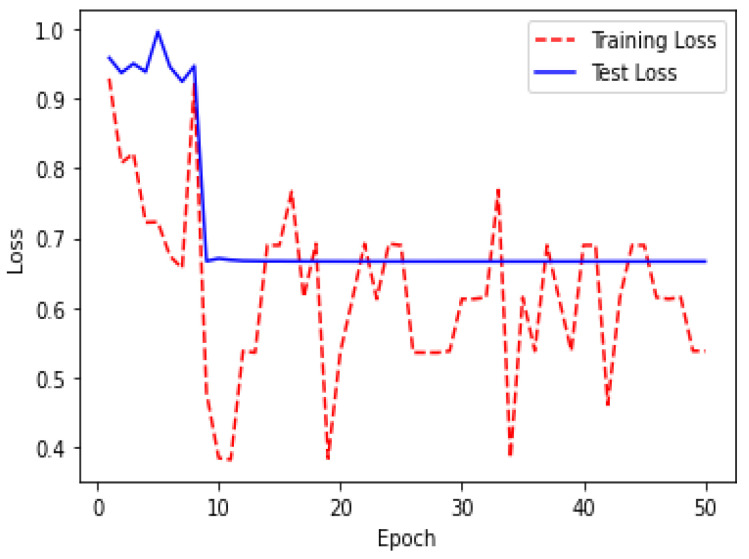
Loss over 50 epochs during the model ResUNet’s training progress for tumor segmentation intervals.

**Figure 19 bioengineering-09-00368-f019:**
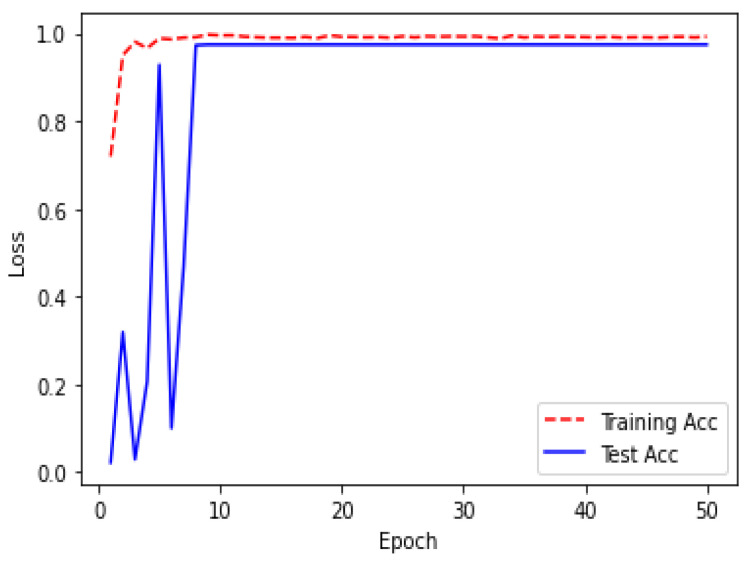
Acc over 50 epochs during the model ResUNet’s training progress for liver segmentation intervals.

**Figure 20 bioengineering-09-00368-f020:**
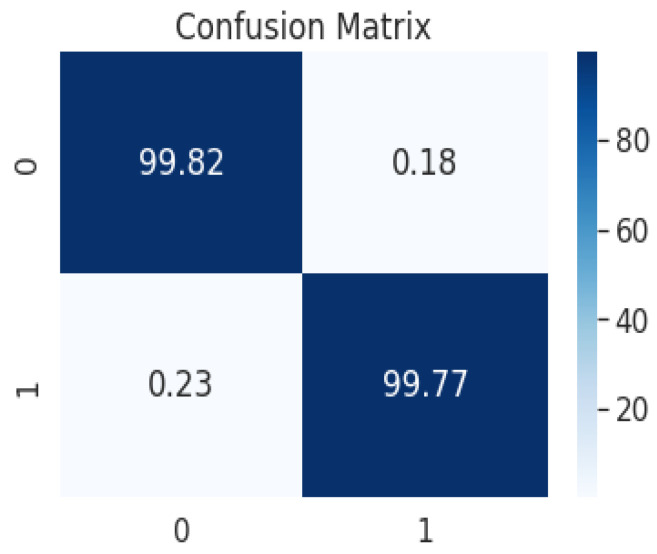
Confusion matrix from the predicted value after tumor segmentation, achieving an Acc of 99.6% and a Dice coefficient of 99.2%.

**Table 1 bioengineering-09-00368-t001:** Loss and Acc results for function values training and validation data on the ResUNet model for liver segmentation using a different number of epochs.

Epoch	Loss	Acc
1	0.3927	0.8608
2	0.4286	0.9696
4	0.4525	0.9867
6	0.5462	0.9593
8	0.4127	0.9794
10	0.2027	0.9673
12	0.6548	0.9632
19	0.3710	0.9776
20	0.4284	0.9923

**Table 2 bioengineering-09-00368-t002:** Loss and Acc results for function values training and validation data on the ResUNet model for Tumor Segmentation using a different number of epochs.

Epoch	Loss	Acc
1	0.2288	0.9196
2	0.5079	0.9504
4	0.7225	0.9660
6	0.6742	0.9864
8	0.8204	0.9913
10	0.3850	0.9953
12	0.4382	0.9924
49	0.5383	0.9906
50	0.2382	0.9927

**Table 3 bioengineering-09-00368-t003:** Segmented results of proposed framework represented as mean ± standard deviations.

Authors	Dice Score	Acc	SVD
[54]	67.5 ± 30.8%	92 ± 3.8%	0.33
[55]	77.11 ± 27.0%	93 ± 3.7%	0.23
[56]	0.58	-	-
[46]	0.63	-	-
Our	99.2%	99.6 ± 3.4%	0.22

## Data Availability

The dataset is available at http://www.ircad.fr/research/3dircadb.

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
