# Peer review of "A Deep Learning Approach for Liver and Tumor Segmentation in CT Images Using ResUNet"

_bioengineering, 2022, doi:10.3390/bioengineering9080368_

Round 1
Reviewer 1 Report
Review of A Deep learning approach for liver and tumor segmentation in CT images using ResUNet
The manuscript submitted by Rahman and co-authors describes a methodology where a U-Net and a ResNet are combined for the segmentation of liver tumours that have been observed in CT datasets. The manuscript is fairly clear in the description but it suffers from some errors and carelessness, it claims to have obtained “great achievements” which are not sustained by evidence, and as such it is not possible to recommend for publication.
Specific comments:
1) The manuscript claims “Performed various preprocessing and data augmentation steps” as a great achievement. This is standard processing of any algorithm
2) The manuscript claims “Achieved 95% mIOU” yet when it presents the metrics used in section 4.2 it only shows sensitivity, specificity, accuracy and error rate. mIOU is never described. Nowhere in the document is the 95% claim sustained.
3) The manuscript is not comparing against literature that have used the same datasets, e.g. https://doi.org/10.1016/j.compbiomed.2017.01.009
4) The confusion matrix shown in Figure 20 indicates that there are far more cases in which the prediction is incorrect than correct.
5) Figure 16 shows a graph for loss but the caption (very badly written) mentions Acc.
6) Some images are not relevant nor interesting, e.g. Fig 4 only shows a histogram. Caption writes “ater” which does not exist in English language. These errors are common along the manuscript.
Author Response
Dear Reviewer
Thank you for reviewing our manuscript and suggesting valuable changes in order to improve the quality of our paper. We have tried our level best to incorporate all the suggested changes as per your suggestions.
Reviewer#1, Concern # 1: The manuscript claims “Performed various preprocessing and data augmentation steps” as a great achievement. This is standard processing of any algorithm
Author response: Thank you for your comments. The proposed ensemble of ResNet and UNet has been implemented on both the original images and preprocessed CT images. The results suggest that the proposed model with preprocessed images has shown better results. Similarly in the case of data augmentation, which is the best technique to cater with fewer amounts of data.
Reviewer#1, Concern # 2: The manuscript claims “Achieved 95% mIOU” yet when it presents the metrics used in section 4.2 it only shows sensitivity, specificity, accuracy, and error rate. mIOU is never described. Nowhere in the document is the 95% claim sustained.
Author response: Thank you for your observation. We have updated the manuscript by adding the detail of ‘Intersection over junction (IOU)” and mIOU in the text and also added the equation (Eq-5).
Author Action: Updated the manuscript with sufficient information related to mIOU and equation-5.
Reviewer#1, Concern # 3: The manuscript is not comparing against literature that have used the same datasets, e.g. https://doi.org/10.1016/j.compbiomed.2017.01.009
Author response: Thank you for your valuable suggestions. Now, we have compared our method with some state-of-the-art methods as mentioned in Table 3.
Author Action: We updated the manuscript by adding the relevant literature and comparison.
Reviewer#1, Concern # 4: The confusion matrix shown in Figure 20 indicates that there are far more cases in which the prediction is incorrect than correct.
Author response: Thank you for your observation. It was added mistakenly, and now we have corrected and added a relevant confusion matrix
Author Action: We have modified Figure 20 by correcting the confusion matrix on page no 16.
Reviewer#1, Concern # 5: Figure 16 shows a graph for loss but the caption (very badly written) mentions Acc.
Author response: Thank you for your observation, again this is a typo error and we have corrected it in the manuscript.
Author Action: We have modified the caption of Figure-16 (During the model ResUNet training Progress for liver segmentation intervals Loss of our dataset for 20 epochs)
Reviewer#1, Concern # 6: Some images are not relevant nor interesting, e.g. Fig 4 only shows a histogram. The caption writes “ater” which does not exist in English language. These errors are common along the manuscript.
Author response: Thanks for your comments. There is a typo mistake in the caption (After’ not ‘ater ’). The histogram equalization is a widely used image contrast enhancement method, and this diagram gives a better understanding to readers that how CT images after contrast adjustment are linearized.
Author Action: Updated in the Fig4 caption on Page no.06.

Reviewer 2 Report
The topic of the article is interesting. The scientific and practical purpose of the article is clearly indicated. The authors described the problem of liver tumor segmentation from the entire organ. The article presents a method of using the ResUNet model for liver and tumor segmentation in CT images. The literature review has been provided in a fair and complete manner. The authors also indicated the limitation of their model and the possibilities of future work. There are several inaccuracies in the article that affect the understanding and quality of the article. I indicate them below.1. Data used in the experiment is described in subsection 3.1. It is not known how many CT scan of patients have been used. The number of images is important when using CNN. The dataset should be large enough for the network to be properly trained. You should also explain the name of the database and note that the authors used the data available on the Internet and did not collect it on their own.
2. The quality of the figures is not very high. Mainly Figure 14, which shows the network architecture, needs to be improved. This is of great importance in understanding the model used.
3. There is no information in the table captions which one represents the results for 20 and 50 epochs. This information is included in the text, but should also be included in the table caption.
4. I suggest improving the Confusion Matrix and adding numerical information on how many specific cases have been classified correctly and how many incorrectly.
5. In section 4.2. Final Results does not have an exact description of the specific numerical results and does not relate them to similar studies available in the literature to demonstrate the clear effectiveness of its method.
6. After explaining the abbreviation, you can use the abbreviated form or use them alternately. Do not use both forms together several times. For example, in line 35-36 computed tomography (CT), the authors introduce the abbreviated form and in the following passages (e.g. line 92) they again use both forms together.
7. The use of abbreviations in brackets should follow the name to which the abbreviation applies, and not at the end of the sentence as in the case of CNN on line 141.
8. Citation in the wrong places, after the main phrase without any clear reference to the source (line 120).
9. Equation (1): unnecessary signs "_" between words.
10. The sentence begins on line 389 on page 13 and ends on page 15. Do not break sentences in such a way with figures or tables.
11. Composition on pages 14 and 15. Figures and tables should be interleaved with text.
12. Unnecessary use of capital letters in the middle of a sentence (e.g. 383, 369, 374, 376)
13. The description of Fig. 18 and Fig. 19 are overlapping.
14. Dividing one word with a picture on lines 186 and 187
I would like to pay attention to the necessity to work on the text. It's just that the article was written not carefully. I recommend improving the stylistics of the text and linguistic corrections.
Author Response
Dear Reviewer
Thank you for reviewing our manuscript and suggesting valuable changes in order to improve the quality of our paper. We have tried our level best to incorporate all the suggested changes as per your suggestions.
Reviewer#2, Concern # 1: Data used in the experiment is described in subsection 3.1. It is not known how many CT scan of patients have been used. The number of images is important when using CNN. The dataset should be large enough for the network to be properly trained. You should also explain the name of the database and note that the authors used the data available on the Internet and did not collect it on their own.
Author response: Thank you for your comments. We have updated as per the reviewer’s suggestions. This looks like:
The 3D-IRCADb-1 (http://www.ircad.fr/research/3dircadb) dataset consists of three-dimensional (3D) CT images of patients, well ordered and publicly available by the IRCAD. Each image has a width and height of 512 * 512 pixels. The depth of each patient’s slice, or the number of slices, varies between 74 and 260, or overall 2,800 slices. That is self-possessed 3D CT scans of 10 men and 10 women within 75% of the cases. Such as DICOM-for- matted patient images, labeled images and mask images are provided as actual data for the segmentation process in preprocessing section. Couinaud segmentation [24] reveals the location of tumor volumes. This illustrates the key challenges of using software to segment the liver [25]
Author Action: We updated the details of the dataset in Section 3.1.
Reviewer#2, Concern # 2: The quality of the figures is not very high. Mainly Figure 14, which shows the network architecture, needs to be improved. This is of great importance in understanding the model used.
Author response: Thank you for your valuable suggestions. We have updated Figure 14 and some other blurry images as per the reviewer’s suggestions and now they are of good quality.
Author Action: We updated the manuscript by updating Figure-14 and set all other images to 300dpi to make them apparent.
Reviewer#2, Concern # 3: There is no information in the table captions which one represents the results for 20 and 50 epochs. This information is included in the text, but should also be included in the table caption.
Author response: We very much appreciate your valuable inputs. We have updated the table captions as per the reviewer’s suggestions. The loss and the accuracy values for liver segmentation is given in Table 1, while the loss and accuracy values for tumor segmentation is mentioned in Table 2.
Author Action: Thanks. We have modified Table 1 and Table 2 on page no 14 and 15.
Reviewer#2, Concern # 4: I suggest improving the Confusion Matrix and adding numerical information on how many specific cases have been classified correctly and how many incorrectly.
Author response: Thanks for your suggestion. We have modified the confusion matrix with numerical information.
Author Action: Details added as Confusion Matrix from the predicted value after tumor segmentation, achieved an Acc of 99.6% and a Dice Coefficient of 99.2% “.
Reviewer#2, Concern # 5: In section 4.2. Final Results does not have an exact description of the specific numerical results and does not relate them to similar studies available in the literature to demonstrate the clear effectiveness of its method.
Author response: Thanks for your comments. We added some extra details of the proposed liver and tumor segmentation based on ResUNet, mIOU, and Table-3 description. Now, it is quite obvious the comparison of the proposed method with some state-of-the-art methods.
Author Action: We have updated the manuscript by adding relevant information in Section 4.2 (Final Results).
Reviewer#2, Concern # 6: After explaining the abbreviation, you can use the abbreviated form or use them alternately. Do not use both forms together several times. For example, in line 35-36 computed tomography (CT), the authors introduce the abbreviated form and in the following passages (e.g. line 92) they again use both forms together.
Author response: Thanks for the comment. We have corrected all such mistakes, and now all such abbreviations are written properly and defined at their first appearances.
Author Action: We updated the manuscript by correcting all such mistakes as on line 94.
Reviewer#2, Concern # 7: The use of abbreviations in brackets should follow the name to which the abbreviation applies, and not at the end of the sentence as in the case of CNN on line 141.
Author response: Thank you for highlighting such mistakes. Yes, this is a typo mistake and we have corrected it as per the reviewer’s suggestion.
Author Action: We updated the manuscript by correcting all such mistakes as on line 144.
Reviewer#2, Concern # 8: Citation in the wrong places, after the main phrase without any clear reference to the source (line 120).
Author response: We very much appreciate your comments. Yes, the reviewer is correct and we have updated as per the reviewer’s suggestions.
Author Action: We updated the manuscript by correcting all such mistakes as on line 122.
Reviewer#2, Concern # 9: Equation (1): unnecessary signs "_" between words.
Author response: Thank you for your observation. We have corrected this equation. Now it looks like:
Author Action: Updated the manuscript by correcting Equation (1)
Reviewer#2, Concern # 10: The sentence begins on line 389 on page 13 and ends on page 15. Do not break sentences in such a way with figures or tables.
Author response: We very much appreciate your comments. We have corrected as per the best of your’s suggestion as mentioned in Line 400 to Line 417.
Author Action: Updated the manuscript as depicted in Line 400-417.
Reviewer#2, Concern # 11: Composition on pages 14 and 15. Figures and tables should be interleaved with text.
Author response: Thank you for your comments. We have updated as per the reviewer’s suggestions.
Author Action: Now all the figures and tables are interleaved with relevant text.
Reviewer#2, Concern # 12: Unnecessary use of capital letters in the middle of a sentence (e.g. 383, 369, 374, 376)
Author response: Thank you for your valuable input. Now all the abbreviations are defined properly at their first appearances.
Author Action: Updated the manuscript by writing in small letters other than abbreviations and common nouns.
Reviewer#2, Concern # 13: The description of Fig. 18 and Fig. 19 are overlapping.
Author response: Thank you for your comments. Fig. 18 represents the liver segmentation intervals loss of our dataset for 50 epochs, while the liver segmentation intervals accuracy of our dataset for 50 epochs is mentioned in Fig. 19.
Author Action: The overlapping details are corrected in Fig.18 and Fig. 19.
Reviewer#2, Concern # 14:
Dividing one word with a picture on lines 186 and 187
Author response: Thank you for your comments. We have updated as per the reviewer’s suggestions
Author Action: Updated the manuscript as per the given suggestions.

Reviewer 3 Report
The Paper is very interesting however some minor things should be corrected:
1. in the introduction set keywords with full name and alphabetically,
2. Figure 1 quality should be improved it looks like its manually streched horizontally.
3. The quality of graph shown in Figure 4 should be improved and the figure should be enlarged. Put the grid in the graph, set labels on x and y axis.
4. Throughout the paper the graph images have inconsistent size of numbers on x and y axis. Please make numbers consistent.
5. The size of figure 14 should somehow be enlarged. There is some text in figure but due to the small size of the figure it is not visible.
6. Graphs shown in figures 16, 17, 18, and 19 should be enlarged, grid must be created, legend should be outside the graph -preferably under the graph. The numbers on x and y axis should be enlarged.
7. Confusion matrix should be enlarged if possible.
Author Response
Dear Reviewer
Thank you for reviewing our manuscript and suggesting valuable changes in order to improve the quality of our paper. We have tried our level best to incorporate all the suggested changes as per your suggestions.
Reviewer#3, Concern # 1: In the introduction set keywords with full name and alphabetically,
Author response: Thank you for your comments. We have changed the keywords alphabetically and in full-form.
Author Action: Updated the keywords section
Reviewer#3, Concern # 2: Figure 1 quality should be improved it looks like it's manually stretched horizontally.
Author response: We very much appreciate your comments. We have restructured Figure-1 and now it is quite understandable.
Author Action: We updated the manuscript by updating Figure 1.
Reviewer#3, Concern # 3: The quality of the graph shown in Figure 4 should be improved and the figure should be enlarged. Put the grid in the graph
Author response: We are very much obliged for highlighting such mistakes. We have improved the figure and also added grid on it.
Author Action: We improve the quality of Figure 4 and enlarged its size as per requirements.
Reviewer#3, Concern # 4: Throughout the paper, the graph images have inconsistent size of numbers on x and y axis. Please make numbers consistent.
Author response: Thanks for the comment. We have refined the image quality and also put all the images in identical format. Now, these images (Fig. 15- Fig- 19 look quite similar in terms of size)
Author Action: Updated the manuscript by adjusting the uniform size of the graphs.
Reviewer#3, Concern # 5: The size of figure 14 should somehow be enlarged. There is some text in the figure but due to the small size of the figure, it is not visible.
Author response: Thank you for your comments. We have corrected this Figure-14 along with the other figures in order to provide a better understanding of the papers to all viewers.
Author Action: We restructured Figure 14 on page no 10.
Reviewer#3, Concern # 6: Graphs shown in figures 16, 17, 18, and 19 should be enlarged, a grid must be created, and legend should be outside the graph -preferably under the graph. The numbers on x and y axis should be enlarged.
Author response: Thanks for your suggestions. We have modified Figures 16, 17, 18, and 19
Author Action: page 14, 15
Reviewer#3, Concern # 7: Confusion matrix should be enlarged if possible.
Author response: Thank you for your comments. We have changed the size of the matrix, and now it is identical in size to other graphs.
Author Action: We updated the size of the confusion matrix on page 15

Round 2
Reviewer 1 Report
I am afraid that the manuscript has done a few cosmetic changes, the authors went for a minimum effort and returned a manuscript which again is full of typos and inaccuracies. I am not going to proof read for them, but just to illustrate my point:
Figure 3. Ater Preprocessing state of image(a) as
I mentioned that the authors were writing ater instead of after and still appears in the document.
There are lots of techniques of medical imagesor each model have their own
benefits and disadvantages.
what about a space between images and or? and lot of techniques of medical images or each model? this is not correct what are they talking about, imaging or processing?
To overwhelmed these issues and recover the excellence of diagnosis of liver cancer, several methods of the calculation have been developed.
This phrase makes absolute no sense. To overwhelm something would be the opposite of trying to solve a problem, and to recover the excellence would imply that there was some excellence that was lost and here the authors claim to be finding it again.
In figure 15, I cannot see how a network would segment what is presented on the right side from the image on the left side. It just does not make any sense.
"Total number of the actual positive" again makes no sense.
the confusion matrix in fig 20 is incorrect as it does not add to 1
Author in reference is is "who", which is clearly incorrect.
Author Response
Dear Reviewer,
Thank you for reviewing our manuscript in order to improve its quality. We do believe that we have incorporated all the suggested changes. Attached is the Response to Reviewer file indicating the response to suggested comments.
Reviewer#1, Concern # 1: Figure 3. Ater Preprocessing state of image(a) as
I mentioned that the authors were writing ater instead of after and still appears in the document.
Author response: Thank you for your comments. We updated the manuscript by incorporating all the suggested changes of grammar, spelling, and punctuation. The Figure caption now looks like:
After preprocessing state of image (a) after HU windowing and (b) using ResUNet Segmentation
Author Action: Updated the manuscript by correcting this spelling mistake.
Reviewer#1, Concern # 2: There are lots of techniques of medical imagesor each model have their own benefits and disadvantages.
what about a space between images and or? and lot of techniques of medical images or each model? this is not correct what are they talking about, imaging or processing?
Author response: Thank you for your observation. This was a typo mistake that was left out unintentionally. We proofread the manuscript to avoid any such dangling modifiers such as:
There are lots of techniques for medical images which have their own advantages and disadvantages.
Author Action: Updated the manuscript by revising this sentence.
Reviewer#1, Concern # 3: To overwhelmed these issues and recover the excellence of diagnosis of liver cancer, several methods of the calculation have been developed.
This phrase makes absolute no sense. To overwhelm something would be the opposite of trying to solve a problem, and to recover the excellence would imply that there was some excellence that was lost and here the authors claim to be finding it again.
Author response: Thank you for your valuable suggestions. The term ‘overwhelmed’ has been replaced with ‘figure out’ which gives proper meaning to this sentence. Also, the whole sentence has been corrected for proper understanding. Now it looks like:
To figure out these issues and to improve the diagnostic performance of liver cancer, several methods of calculation have been developed.
Author Action: We updated the manuscript by adding the relevant literature and comparison.
Reviewer#1, Concern # 4: In figure 15, I cannot see how a network would segment what is presented on the right side from the image on the left side. It just does not make any sense.
Author response: Thank you for your comments. Yes, we do agree that the complete segmentation framework is lacking for liver and liver tumor segmentation in Figure 15. Now, we updated the Fig. 15 and adds the following description in text as well:
Figure 15 presents the implementation framework of deep learning neural network on liver and liver tumor segmentation. It involves image preprocessing steps such as noise reduction, standardization, and normalization techniques to enhance the image quality. Data augmentation operations i.e. reflect the image, rotate the image, and mask are performed to augment the training samples. ResUNet was prepared to find the Region of Interest (ROI) from the nearby organs using CT scans. After separating the ROI and training on CT scans of the liver, the ResUNet was used to segment tumors in the liver.
Author Action: We have modified Figure 20 by correcting the confusion matrix on page no 16.
Reviewer#1, Concern # 5: "Total number of the actual positive" again makes no sense.
Author response: Thank you for your suggestions. Yes, we do agree that there is no need for ‘Total number of actual positive’ in the formula. So we have deleted equation (1) and also corrected other equations too.
Author Action: We have corrected Eq(1) and Eq(2).
Reviewer#1, Concern # 6: the confusion matrix in fig 20 is incorrect as it does not add to 1
Author response: Thanks for your comments. We have considered the concrete values for positive predicated labels initially and now corrected them as per your suggestions as mentioned in Fig. 20.
Author Action: Updated the confusion matrix diagram (Fig 20) with complete count.
Reviewer#1, Concern # 7: Author in reference is is "who", which is clearly incorrect.
Author response: Thanks for your comments. We have corrected reference#1 as given:
World Cancer Report. World Health Organization 2021. Available at https://www.who.int/ 448
news-room/fact-sheets/detail/cancer
Author Action: Corrected reference no. 1.
